# Risk Assessment of the Possible Intermediate Host Role of Pigs for Coronaviruses with a Deep Learning Predictor

**DOI:** 10.3390/v15071556

**Published:** 2023-07-15

**Authors:** Shuyang Jiang, Sen Zhang, Xiaoping Kang, Ye Feng, Yadan Li, Maoshun Nie, Yuchang Li, Yuehong Chen, Shishun Zhao, Tao Jiang, Jing Li

**Affiliations:** 1College of Mathematics, Jilin University, Changchun, Jilin 130012, China; 2State Key Laboratory of Pathogen and Biosecurity, Beijing Institute of Microbiology and Epidemiology, AMMS, Beijing 100071, China

**Keywords:** swine coronavirus, intermediate host, deep learning, dinucleotide composition representation (DCR), adaptation

## Abstract

Swine coronaviruses (CoVs) have been found to cause infection in humans, suggesting that Suiformes might be potential intermediate hosts in CoV transmission from their natural hosts to humans. The present study aims to establish convolutional neural network (CNN) models to predict host adaptation of swine CoVs. Decomposing of each *ORF1ab* and *Spike* sequence was performed with dinucleotide composition representation (DCR) and other traits. The relationship between CoVs from different adaptive hosts was analyzed by unsupervised learning, and CNN models based on DCR of *ORF1ab* and *Spike* were built to predict the host adaptation of swine CoVs. The rationality of the models was verified with phylogenetic analysis. Unsupervised learning showed that there is a multiple host adaptation of different swine CoVs. According to the adaptation prediction of CNN models, swine acute diarrhea syndrome CoV (SADS-CoV) and porcine epidemic diarrhea virus (PEDV) are adapted to Chiroptera, swine transmissible gastroenteritis virus (TGEV) is adapted to Carnivora, porcine hemagglutinating encephalomyelitis (PHEV) might be adapted to Primate, Rodent, and Lagomorpha, and porcine deltacoronavirus (PDCoV) might be adapted to Chiroptera, Artiodactyla, and Carnivora. In summary, the DCR trait has been confirmed to be representative for the CoV genome, and the DCR-based deep learning model works well to assess the adaptation of swine CoVs to other mammals. Suiformes might be intermediate hosts for human CoVs and other mammalian CoVs. The present study provides a novel approach to assess the risk of adaptation and transmission to humans and other mammals of swine CoVs.

## 1. Introduction

The ongoing global COVID-19 pandemic, caused by the pathogenic agent of severe acute respiratory syndrome coronavirus (CoV) 2 (SARS-CoV-2) has still been threatening human health. What is more worrisome is that little is known about how the virus crossed species barriers, adapting to human beings from its natural reservoir host of bat [1,2,3]. A paradigmatic cross-species transmission from natural reservoir host to an intermediate host and then to human beings has been widely accepted for human pandemic viruses, such influenza A viruses (IAVs) [4,5,6] and CoVs [7]. IAVs and CoVs with single-stranded RNA genomes mutate with much higher rates than double-stranded RNA viruses and DNA viruses (single- or double-stranded) [8]. Avian host-originated IAVs of H5N1 [9,10,11], H7N9 [12,13], and other subtypes [14,15] adapted to mammals or mammalian cells with or without mammalian passages. In particular, domestic pigs and probably other species in Suiformes played a key intermediate host role for all six historic influenza pandemics [6,16,17,18,19,20]. Suiformes are also one main family of mammalian hosts for CoVs [21,22,23,24,25,26]. Swine CoVs have been implied to replicate efficiently in human primary cells [27] and, more worryingly, cause infection in malnourished Haitian children [28] or calves and chickens [29]. Thus, it is urgent to pay more attention to the assessment of the adaptation and transmission risk to human beings or other mammals of swine CoVs.

CoVs not only directly infect a variety of mammals in the family of Chiroptera, Artiodactyla, Rodents, Lagomorphs, Carnivores, and Primates, but also cause trans-species infection among these mammals [7,30,31]. Such trans-species infection and transmission have been indicated to play important role for CoVs to adapt to human beings and cause prevalence [7]. Among human CoVs, SARS-CoV-2 is closely related to some bat CoVs [1]. HCoV-OC43 and HCoV-HKU1 originated from cattle [32] and rodents [33] respectively, and HCoV-NL63 originated from bats [34]. SARS-CoVs which originated from bats infected civets before infecting humans and causing an epidemic [35]. In the transmission of HCoV-229E and MERS-CoV from bats to humans, dromedary camels and camelids played the role of intermediate hosts, respectively [7]. Therefore, mammals might serve as intermediate hosts in the transmission of CoVs from their original hosts to humans.

Phylogenetic analysis and other traditional methods in bioinformatics have provided indicative clues to infer the homology to other mammalian CoVs and the potential infection/transmission risk to these hosts of Suiformes CoVs. Phylogenetic trees have indicated that swine CoVs, such as swine acute diarrhea syndrome CoV (SADS-CoV) [21] and porcine epidemic diarrhea virus (PEDV) [25] may originate from bats [26]. Swine transmissible gastroenteritis virus (TGEV) has very high homology to canine CoVs [36]. Results on multiple mammalian cell lines of different species showed susceptibility to SADS-CoV, implying possible multiple host adaptability of Suiformes CoVs [22]. Although biologically experimental results are very reliable, such experiments are time-consuming, resulting in a lag in research results. Furthermore, these methods can hardly quantify and intelligently assess the risk of infection and transmission of Suiformes CoVs to humans and other mammals.

Artificial intelligence (AI) methods have been found to be effective in solving these problems. Sequence compositions of nucleic acids and proteins are significantly associated with genome evolution and adaptation across all kingdoms of life [37]. Adaptive determinants have recently been widely identified at the nucleic acid level (genomic DNA, RNA, or mRNA) among pathogens such as parasites [38], bacteria [39], and viruses [40,41]. The dynamic homeostasis of genomic RNA sequences shapes the transcription, translation, and decay of mRNA [42], particularly for RNA viruses. These determinants regulate the replication of pathogens in hosts via the machinery related to codon usage bias [38,39,43], dinucleotide composition [44], tRNA abundance [39,41], mRNA decay [45], translation elongation speed [46], and translation efficiency [47]. Thus, the RNA sequence-based nucleotide composition is biologically meaningful and is closely related to the causal inference of the virus phenotype. Deep learning based on unlabeled amino acids (AAs) was utilized to extract statistical representation with rich semantics from fundamental features of a protein [48]. Viral escape was modeled using natural language processing (NLP) methods to predict the structural escape patterns [49,50]. Machine learning models based on compositional traits [36,40] such as codon, codon pair, and dinucleotide (DNT) [44,51] have enabled accurate predictions of virus adaptation to hosts. Several models have been utilized to predict SARS-CoV-2 variants on the basis of viral protein sequences, with a particular focus on key mutant AAs related to receptor binding [43,52,53], although easily falling into the trap of overfitting. Convolutional neural networks (CNNs) are widely used in the field of image recognition; however, in recent years, CNN has also performed well in predicting the adaptability of virus hosts. A CNN predictor based on the dinucleotide composition representation (DCR) [54] and AA [55] could provide real-time predictions of emerging SARS-CoV-2 variants. Thus, AI methods are expected to learn the adaptation of swine CoVs to other mammals, and to assess the possible intermediate host role of pigs for coronaviruses.

The present study aims to establish 3D-CNN classification models based on viral genomic DCR to predict the adaptation of Suiformes CoVs to the five types of hosts. The binding ability to specific receptors and the replication ability in host cells are considered key factors influencing viral host adaptation. Therefore, these two major viral proteins—receptor-binding glycoprotein (Gp; also named *S* for CoVs) and RNA-dependent RNA polymerase (RdRp; mainly *ORF1ab*-encoded)—were assessed by the classification traits. After *ORF1ab* and *Spike* genome decomposition by DCR and other traits, unsupervised learning was performed to analyze the distance between different types of Suiformes CoVs and CoVs from other adaptive hosts, and to filter out the traits with better interpretation of CoV sequences. On this basis, CNN models with five adaptation labels of *ORF1ab* and *Spike* were established to predict the host adaptation of each Suiformes CoV. The spatiotemporal distribution of the adaptation ratio was predicted by the models and through descriptive statistics. The rationality of the models was verified by establishment of phylogenetic trees. Our study predicts the adaptability of Suiformes CoVs to five types of hosts, and scientifically assesses the infection risk of existing or novel Suiformes CoVs to various mammalian hosts, especially humans.

## 2. Methods

### 2.1. Data Processing, Host Labeling, and Sequence Decomposition of CoVs

*ORF1ab* and *Spike* sequences were selected from full genome sequences of CoVs downloaded from NCBI nucleotide database (https://www.ncbi.nlm.nih.gov/nuccore) (accessed on 31 December 2019) after data cleaning. SARS-CoV-2 samples were downloaded from the GISAID CoV database (https://www.epicov.org/epi3/frontend) (accessed on 30 June 2021) [56], and then randomly sampled. Each CoV sample was marked with a collection date and continent, and then labeled according to the order or suborder of its adaptive hosts. Adaptive hosts of the samples were divided into Chiroptera (CHI), Artiodactyla (ART) (not including Suiformes CoVs), Rodent and Lagomorpha (ROD_LAG), Carnivora (CAR), Primates (PRI), and Suiformes (SUI). SUI CoV samples were further labeled with SADS-CoV, PEDV, PHEV, TGEV, and Porcine deltacoronavirus (PDCoV). A nucleotide counting script of python was utilized for genome sequence decomposition [54]. The frequency of six types of compositional traits (20 amino acids (AAs), 12 nucleotides (NTs), 48 dinucleotides (DNTs), 64 codons, 1536 dinucleotide composition representations (DCRs), and 3721 codon pairs were counted for *ORF1ab* or *Spike* sequence of each CoV sample with the following formula, where ‘count’ represents the quantity, and ‘seq_len’ represents the total length of the selected gene sequence.
$$Freq\left(NT\right)=count\left(NT\right)\times 4\times \frac{3}{seq\_len},$$
$$Freq\left(DNT\right)=count\left(DNT\right)\times 16\times \frac{3}{seq\_len},$$
$$Freq\left(DCR\right)=count\left(DCR\right)\times 256\times \frac{3}{seq\_len-1},$$
$$Freq\left(Codon\right)=count\left(Codon\right)\times 64\times \frac{3}{seq\_len},$$
$$Freq\left(Codon\;pair\right)=count\left(Codon\;pair\right)\times 3721\times \frac{3}{seq\_len-3},$$
$$Freq\left(AA\right)=count\left(AA\right)\times 20\times \frac{3}{seq\_len}.$$

### 2.2. Reduction, Visualization, and Clustering of Six Types of Compositional Traits of ORF1ab and Spike Sequences of CoVs

Considering the diversity of CHI CoVs, CHI CoV samples were screened on the basis of the two main components reduced from DCR, so as to eliminate some abnormal samples. After screening of the CHI CoVs, there were 301 CHI samples, 189 related to SARS-CoV and the remaining 112 related to other CHI CoVs. In the subsequent sampling, in order not to disrupt the sample distribution of CHI, these two types of CHI CoVs were sampled separately. The sample sizes for ART, ROD_LAG, CAR, PRI, and SUI (including 41 SADS-CoV samples, 530 PEDV samples, 13 PHEV samples, 58 TGEV samples, and 40 PDCoV samples) were 579, 48, 79, 485, and 682, respectively. Down-sampling (using the pandas.DataFrame.sample of python) was conducted to reduce the sample size of SUI CoVs to 65 (SADS-CoV, PEDV, PHEV, TGEV, and PDCoV each accounted for 20.0%). Due to the small number of PHEV samples in SUI CoVs, oversampling was not performed in order to prevent the generated data from deviating from the original distribution. Down-sampling was also performed to reduce sample size of CHI, ART, and PRI CoVs to about 97, and over-sampling (using imblearn.over_sampling.SMOTE of python) was also conducted to ensure that the numbers of samples from CHI, ART, ROD_LAG, CAR, and PRI were identical. After sampling, there were about 550 samples in total (CHI, ART, ROD_LAG, CAR and PRI CoVs each accounted for 17.6%, and SUI CoVs accounted for 11.8%).

The sklearn.decomposition.PCA and sklearn.manifold.TSNE packages (https://scikit-learn.org/stable/about.html#citing-scikit-learn) were utilized to perform t-distributed stochastic neighbor embedding (t-SNE) and principal component analysis (PCA) for dimensional reduction of six types of compositional traits of *ORF1ab* or *Spike* of five types of SUI CoVs and other CoVs, as well as their visualization. Two main components reduced from compositional traits by t-SNE or PCA were extracted. t-SNE and PCA were used to test whether samples from different adaptive hosts can be distinguished on the basis of various composition traits, and to observe the relationship among five types of SUI CoVs and CoVs from other adaptive hosts. To further study the distance and clustering five types of SUI CoVs and other CoVs, hierarchical clustering of each type of composition trait of *ORF1ab* and *Spike* was conducted using the sns.clustermap of python. In order to prevent samples from being hidden in hierarchical clustering results, fewer samples were further extracted from various types of CoVs. Down-sampling was performed to reduce samples of CHI, ART, ROD_LAG, CAR, and PRI to 20 each, and samples of SUI to 65 (SADS-CoV, PEDV, PHEV, TGEV, and PDCoV each accounted for 20.0%). After sampling, there were 165 samples in total (CHI, ART, ROD_LAG, CAR, and PRI CoVs each accounted for 12.1%, and SUI CoVs accounted for 39.4%).

### 2.3. Establishment of Convolutional Neural Network (CNN) Models of ORF1ab and Spike Sequences

Unsupervised learning was performed so as to select the compositional traits able to distinguish the samples from CHI, ART, ROD_LAG, CAR, and PRI, and to analyze the relationship among five types of Suiformes CoVs and samples from other adaptive hosts. On the basis of the results of unsupervised learning, 1536 DCRs were selected for supervised learning. The samples were labeled on the basis of their adaptation hosts. The adaptation labels of 0, 1, 2, 3, and 4 indicated adaptation to CHI, ART, ROD_LAG, CAR, and PRI. Down-sampling and over-sampling were performed to keep the number of samples from CHI, ART, ROD_LAG, CAR, and PRI balanced. Sampling in supervised learning was consistent with sampling through dimensionality reduction of unsupervised learning. Specifically, 25% of *ORF1ab* and *Spike* sequences of the samples other than SUI CoVs were randomly sampled for the test set, and the remaining sequences were used to train the 3D-CNN classification models based on the *ORF1ab* and *Spike* sequences. The adaptation of *ORF1ab* and *Spike* sequences of SUI CoV samples was predicted by the trained models. The 1536-dimensional DCR of *ORF1ab* and *Spike* sequences was respectively reshaped into an array of (6, 16, 16). Three-layer CNN models were established with a kernel size of convolution of (1, 3, 3), kernel size of average pooling of (1, 2, 2), stride of (1, 1, 1), padding of (0, 1, 1), and learning rate of 0.001. The ReLU and Sigmoid functions were selected for activation. The softmax function was used to output the adaptive probability for five types of hosts. The host with the maximum prediction value was considered to be the adapted host. Average pooling was selected for the models. The predictive effect of the CNN classification models was confirmed by plotting the training loss of the models with epochs from one to 50, as well as the confusion matrix, the receiver operating characteristic curve (ROC) and the area under curve (AUC), and pair-plotting of PCA1 and PCA2 of the fully connected layers (FC) data with epochs of 10, 20, 30, 40, and 50 of the *ORF1ab* and *Spike* models. Models with a training epoch of 50 were finally selected.
$$Relu\;function:ReLu\left(x\right)=\max\left(0,\;x\right),$$
$$Sigmoid\;function:Sigmoid\left(x\right)=\frac{1}{1+e^{-x}},$$
$$Softmax\;function:Softmax\left(z_{i}\right)=\frac{e^{z_{i}}}{\sum_{c=1}^{C}e^{z_{c}}}.$$

### 2.4. Prediction of the Adaptation Ratio of Suiformes CoVs to Various Hosts and the Spatiotemporal Distribution of the Adaptation Ratio

In order to assess and visualize the adaptation of SUI CoVs to CHI, ART, ROD_LAG, CAR, and PRI, the adaptation ratio of SUI CoVs to these hosts was plotted according to the predicted results of both *ORF1ab* and *Spike* sequences. The temporal and spatial distributions of the adaptation ratios were also calculated via descriptive statistics. The SUI CoV samples were classified on the basis of the collection year and collection continent, and the adaptation ratios of SUI CoV samples to various hosts were also temporally and spatially plotted according to the predicted labels of *ORF1ab* and *Spike* sequences. The host adaptation of each type of SUI CoVs was also predicted.

### 2.5. Phylogenetic Analysis of ORF1ab and Spike Genes

In order to explore the phylogenetic relationship of the samples originating from PHEV, rabbitHKU14, and HCoV-OC43, phylogenetic trees were constructed on the basis of *ORF1ab* and *Spike* genes. The amino-acid sequences of all 12 PHEV and five rabbitHKU14 samples with known collection information, and the randomly sampled 15 HCoV-OC43 samples obtained from Section 2.1 were first aligned by MAFFT [57], and maximum likelihood (ML) trees were constructed using RAxML v8.2.12 [58] with 100 bootstrap iterations and other variables set to default. Phylogenetic trees were visualized using iTol [59]. Furthermore, in order to explore the phylogenetic relationship of CoVs originating from all species, proportional random sampling was carried out according to the number of samples. Some samples were first randomly sampled from the CoV samples with known collection information to constructed trees. On this basis, according to the genetic distance between samples on the tree, a portion of the representative samples (99 samples in total) were selected to rebuild the tree. The subsequent process was as described above.

### 2.6. Biolayer Interferometry (BLI) Assay for RBD of PHE-CoV Spike and NCAM Interaction

BLI binding experiments were performed in a Gator™ Label-Free Bioanalysis System (Gator Bio). The concentration of NCAM protein bound to the Anti-His probe was 5 µg/mL, while the *Spike* protein was double-gradient-diluted from 80 µg/mL to 1.25 µg/mL. Binding sensorgrams were aligned to dissociation, following subtraction of the reference well/sample, and globally fit to a 1:1 binding model. The equilibrium dissociation constant (KD) was calculated using the instrument’s software and visualized using GraphPad Prism 9.0 software.

The recombinant rat (80399-R08H) and human (10673-H08H) NCAM protein were purchased from Sino Biological. The rabbit NCAM protein (XP_051705223.1) and RBD of the PHEV *Spike* protein were expressed and purified by Sino Biological.

## 3. Results

### 3.1. Prediction Pipeline of Adaptive Hosts of SUI CoVs

The downloaded CoV samples were labeled with six hosts, namely, CHI, ART, ROD_LAG, CAR, PRI, and SUI. Among them, SUI CoV samples included SADS-CoV, PEDV, PHEV, TGEV, and PDCoV. The *ORF1ab* and *Spike* sequences of each sample were extracted and decomposed with six types of compositional traits, namely, 20 AAs, 12 NTs, 64 codons, 3721 codon pairs, 48 DNTs, and 1536 DCRs (Figure 1A). t-SNE, PCA, and hierarchical clustering were conducted for the reduction in and visualization of these traits, so as to analyze the distance between SUI CoVs and samples from other adaptive hosts of each compositional trait (Figure 1B). Deep learning based on DCR was performed to establish classification models for adaptive prediction of SUI CoVs. Models based on *ORF1ab* or *Spike* sequences were established (Figure 1C). Prediction of the adaptive hosts of each type of SUI CoV and the temporal and spatial distributions of the adaptation ratio were obtained using the models (Figure 1D).

### 3.2. Unsupervised Learning of SUI and Other CoVs

t-SNE and PCA were conducted for visualization and dimensional reduction of each type of compositional trait of *ORF1ab* and *Spike* sequences of CoVs. The samples from CHI, ART, ROD_LAG, CAR, and PRI were significantly separated on the basis of two main components reduced by both t-SNE (Figure 2A) and PCA (Figure 2B) of the 1536-dimentional-DCR of *ORF1ab* sequences. The sample size of ROD_LAG CoVs was relatively small, and some of these samples were generated through over-sampling; therefore, ROD_LAG CoV samples were relatively scattered in the dimensionality reduction result. According to the dimensional reduction results, SADS-CoV and PEDV gathered with the CoVs from CHI, PHEV gathered with the CoV samples from PRI and ROD_LAG, TGEV gathered with CAR CoV samples, and PDCoV gathered with CoVs from CHI based on the DCR of *ORF1ab* and CAR based on the DCR of *Spike* (Figure 2A,D). The distance between PDCoV and ART samples was relatively small for *ORF1ab*, and the distance between PDCoV and CHI samples was also relatively small for *Spike* (Figure 2B,E), suggesting that there might also be some homology among these gene sequences. The relationship between the samples obtained by hierarchical clustering based on the DCR of *ORF1ab* and *Spike* was similar to the distribution obtained by reduction (Figure 2C,F). t-SNE, PCA, and hierarchical clustering based on AAs (Appendix A), NTs (Appendix A), DNTs (Appendix A), codons (Appendix A), and codon pairs (Appendix A) indicated similar clustering or separation of these CoVs. DCR performed well in separating samples from various adaptive hosts. It can also be inferred that the distance among different types of Suiformes CoVs and CoVs from other hosts is different, making it necessary to establish new methods to predict and identify the host adaptability of each type of SUI CoVs.

### 3.3. Classification Effect of the DCR-Based CNN Models of ORF1ab and Spike

According to the distance between CoVs of various hosts, DCR-based CNN classification models were built to predict the adaptation of *ORF1ab* and *Spike* of SUI CoVs to different types of hosts. For both the *ORF1ab* (Figure 3A) and the *Spike* (Figure 3B) models, the training loss was relatively high with an epoch of 10. When the epoch value increased, the training loss decreased significantly, and the training loss was very close to 0 with an epoch of 50. The low values in the confusion matrices (Appendix A and Figure 3C), as well as the ROC and AUC (Appendix A and Figure 3D), showed the low accuracy of prediction of *ORF1ab* and *Spike* models with an epoch value of 10. The separation of samples with adaptation to CHI and those with adaptation to ART was not clearly indicated in the pair-plot of PCA1 and PCA2 of the FC data of *Spike* gene model (Figure 3E and Appendix A). The higher prediction accuracy was reflected in the confusion matrices (Appendix A and Figure 3F) and ROC and AUC (Appendix A and Figure 3G) of *ORF1ab* and *Spike* models with an epoch of 30. However, the pair-plotting of the *Spike* gene model was not able to distinguish the samples with adaptation to ART and samples with adaptation to CAR (Figure 3H and Appendix A). The pair-plotting of *ORF1ab* model with epochs of 10 and 30 indicated relatively clear separation of samples from different types of hosts (Appendix A). The high accuracy and low training loss value showed that the epoch of 50 should be selected for both *ORF1ab* and *Spike* models (Appendix A and Figure 3A,B,I,J). Separation of samples from various adaptive hosts was also illustrated in the pair-plotting of *ORF1ab* (Appendix A) and *Spike* (Figure 3K and Appendix A) models with an epoch of 50. The performances of both *ORF1ab* (Appendix A–F,J–L) and *Spike* (Appendix A–F,J–L) models with an epoch of 20 and 40 were also inferior to those with epoch of 50. Additionally, MERS-CoV is closely related to bat CoV HKU5, resulting in a relatively low prediction accuracy for samples adapted to ART (Appendix A). There are few CoVs adapted to ROD_LAG; therefore, many samples were obtained through over-sampling, which makes the distinction between these samples and the samples adapted to other hosts not very significant (Appendix A and Figure 3K).

### 3.4. Adaptation Prediction of Suiformes CoVs Based on the CNN Models

The adaptation of SUI CoVs to CHI, ART, ROD_LAG, CAR, and PRI was predicted by the DCR-based CNN models of *ORF1ab* and *Spike* sequences. The prediction results showed that, among the SUI CoV samples, 85.8% were predicted to be adaptive to CHI, along with 3.8% to ART, 8.5% to CAR, and 1.9% to PRI for *ORF1ab*, and 86.0% to CHI, 1.9% to ROD_LAG, and 12.0% to CAR for *Spike* (Figure 4A,D).

The spatiotemporal distribution of the adaptation ratio of SUI CoVs to five types of hosts was also obtained. Specifically, 16.7% of the SUI CoV samples collected from 1952 to 2000, 29.4% of the samples collected from 2001 to 2010, 89.2% of the samples collected from 2011 to 2021, and 83.3% of the samples with unknown collection date showed adaptation to CHI, 4.5% of the SUI CoV samples collected from 2011 to 2021 showed adaptation to ART, 83.3% of the SUI CoV samples collected from 1952 to 2000, 64.7% of the samples collected from 2001 to 2010, 4.4% of the samples collected from 2011 to 2021, and 15.4% of the samples with unknown collection date showed adaptation to CAR, and 5.9% of the samples collected from 2001 to 2010, 1.9% of the samples collected from 2011 to 2021, and 1.3% of the samples with unknown collection date showed adaptation to PRI for *ORF1ab*. On the other hand, 16.7% of the SUI CoV samples collected from 1952 to 2000, 29.4% of the samples collected from 2001 to 2010, 89.5% of the samples collected from 2011 to 2021, and 83.3% of the samples with unknown collection date showed adaptation to CHI, 5.9% of the samples collected from 2001 to 2010, 1.9% of the samples collected from 2011 to 2021, and 1.3% of the samples with unknown collection date showed adaptation to ROD_LAG, and 83.3% of the samples collected from 1952 to 2000, 64.7% of the samples collected from 2001 to 2010, 8.5% of the samples collected from 2011 to 2021, and 15.4% of the samples with unknown collection date showed adaptation to CAR for *Spike* (Figure 4B,E). Furthermore, 78.9% of the SUI CoV samples collected from Europe, 85.2% of the samples collected from North America, 85.6% of the samples collected from Asia, and 92.3% of the samples with unknown collection continent were predicted to be adaptive to CHI, 1.1% of the samples collected from North America and 7.0% of the samples collected from Asia were predicted to be adaptive to ART, 15.8% of the SUI CoV samples collected from Europe, 10.2% of the samples collected from North America, 6.7% of the samples collected from Asia, and 7.7% of the samples with unknown collection continent were predicted to be adaptive to CAR, and 5.3% of the SUI CoV samples collected from Europe, 3.5% of the samples collected from North America, and 0.6% of the samples collected from Asia were predicted to be adaptive to PRI for *ORF1ab*. On the other hand, 78.9% of the SUI CoV samples collected from Europe, 82.0% of the samples collected from North America, 89.0% of the samples collected from Asia, and 92.3% of the samples with unknown collection continent were predicted to be adaptive to CHI, 5.3% of the SUI CoV samples collected from Europe, 3.5% of the samples collected from North America, and 0.6% of the samples collected from Asia were predicted to be adaptive to ROD_LAG, and 15.8% of the SUI CoV samples collected from Europe, 14.5% of the samples collected from North America, 10.4% of the samples collected from Asia, and 7.7% of the samples with unknown collection continent were predicted to be adaptive to CAR for *Spike* (Figure 4C,F).

### 3.5. Adaptation Prediction, Phylogenetic Analysis, and Receptor Binding Verification of PHEV

The host adaptation of each type of SUI CoVs was predicted in detail using the deep learning models with five adaptation labels of both *ORF1ab* and *Spike* sequences. Specifically, 100% of *ORF1ab* and *Spike* genes were predicted to be adaptive to CHI for SADS-CoVs and PEDVs, or to CAR for TGEVs (Figure 5A,B). However, 100% of PHEVs were predicted to be adaptive to PRI for *ORF1ab* and to ROD_LAG for *Spike* genes. Additionally, 66.7% of PDCoV was predicted to be adaptive to ART, and the remaining 33.3% was predicted to be adaptive to CHI for *ORF1ab*, while 61.5% of PDCoV was predicted to be adaptive to CAR, and the remaining 38.5% was predicted to be adaptive to CHI for *Spike* (Figure 5A,B). When both the *ORF1ab* and the *Spike* sequences of the Suiformes CoV samples are predicted to be adaptive to the same type of hosts, it can be inferred that the samples have significant adaptability to this type of host; otherwise, if the two genes of the samples are predicted to be adaptive to different types of hosts, the samples may have certain adaptability to those types of hosts. In order to elucidate the phylogenetic relationships between PHEVs and ROD_LAG or PRI CoV samples, two phylogeny trees were constructed on the basis of rgw *ORF1ab* and *Spike* genes of PHEV, rabbit CoVs (HKU14), and HCoV-OC43, with the closest distance in DCR traits. Interestingly, PHEV *ORF1ab* genes were also closer in phylogenetic relationship with those of HCoV-OC43 (Figure 5C), while *Spike* genes were in an independent branch from rabbit HKU14 CoVs and HCoV-OC43, relatively closer to the former (Figure 5D). These results were consistent with the predicted adaptation with the CNN model, confirming the accuracy of the adaptation prediction with deep learning methods. Then, after random sampling of CoV sequences from all five species, we similarly constructed another two phylogeny trees on the basis of ORF1ab and *Spike* genes. The results showed that the genetic distances of CoVs from the same species were not necessarily close in the phylogenetic trees, and some of them were even far away (Appendix A). This signified that our prediction model effectively complemented the traditional phylogenetic relationship, making it suitable for extensive application.

To further verify the analysis results of our models, we tried to explore the interaction between the PHEV *Spike* protein and cell lines of human, rat, and rabbit. According to previous studies, we determined that a fragment (258-amino-acid fragment, residues 291–548) located in *Spike* (277–794 is the RBD of the PHEV *Spike* protein) and a cellular neural cell adhesion molecule (NCAM) were the key receptor proteins [60,61]. Then, BLI binding experiments were performed to measure the binding ability of PHEV *Spike* RBD to NCAM of different host cells. We found that the PHEV *Spike* RBD could bind weakly to rabbit NCAM protein, with a K_D_ of 5.73 × 10^−11^ M, whereas it could hardly bind to human or rat NCAM protein (Figure 5E,F and Appendix A). This means that the PHEV *Spike* protein was more adaptive to rabbit, rather than rat or human, which was basically consistent with our prediction results that 100% of PHEV samples were predicted to be adaptive to ROD_LAG for *Spike* genes (Figure 5B). However, our prediction model could be further optimized, e.g., through refinement of host classification. Taken together, the DCR-based CNN models could accurately predict the host adaptability of SUI CoVs, and the results were basically consistent with the phylogenetic relationship and experiment results.

## 4. Discussion

This study evaluated the risk of SUI CoVs to infect and transmit in humans and other types of hosts, shedding light on whether SUI CoVs can spread across species, or whether pigs might be a possible intermediate host for CHI CoVs. Genomic DCR traits were qualified to distinguish the CoVs from various hosts. SUI SADS-CoV and PEDV samples were predicted to be adapted to CHI, and TGEV samples were predicted to be adapted to CAR according to DCR-based deep learning models with five adaptation labels for both *ORF1ab* and *Spike*. PHEV samples were predicted to be adapted to PRI according to *ORF1ab* and ROD_LAG according to *Spike*. Suiformes acted as intermediate hosts in H1N1 transmission [62], and it is predicted that SUI CoVs are also able to adapt to various hosts; therefore, they might also be intermediate hosts of various CoVs. PHEV is phylogenetically closely related to HCoV-OC43 [63] and is predicted to be adaptive to the PRI-based genomic DCR of *ORF1ab*. Therefore, there is a possibility that Suiformes can be intermediate hosts during the transmission of CoVs to humans.

Worryingly, it had been recently found that some strains of PDCoV are able to infect humans [28], implying evidence that pigs might be potential intermediate hosts for CoVs from CHI or other mammalian hosts. In order to further assess the risk of infection and transmission of PDCoV in the population, the host adaptation of the human-infected PDCoV was predicted using our deep learning models based on genomic DCR. It was found that these PDCoVs had a low probability of adapting to PRI, showing that, although Suiformes are potential intermediate hosts of PDCoV, these CoV strains may only cause spillover human infection in Haitian malnourished children, being unable to cause large-scale infection and epidemic in the human population. The existing studies have shown that SADS-CoV [23] and PEDV [25] originate from bat, and TGEV and PDCoV are respectively closely related to some CAR CoVs [36] and avian hosts [64] by comparing the homology of the sequences, which proves the rationality of our prediction. Compared with the existing research, the present method can quantify and compare the adaptability to different types of hosts in a timely manner.

Machine learning has been utilized to investigate the host adaptability of coronaviruses [65]. Liam et al. trained random forest models independently on genome composition biases of *Spike* protein and whole-genome sequences, including dinucleotide and codon usage biases in order to predict animal host. The combination of genetic resources with machine learning algorithms was consistent with our purpose; however, different modeling methods were used in our research. Moreover, codon usage and amino-acid preferences are other important factors that may influence the host adaptability of virus [66,67]. In essence, codon usage and amino-acid preferences are also based on nucleotide composition, which successfully explain the evolution of PEDV under the mutation pressure and natural selection. In our study, machine learning models provided a more systematic explanation for genome composition analysis and predicted the host adaptability of the virus from another point.

The binding ability of CoV Spike protein to a specific receptor can directly reflect the host adaptability of virus. To further verify this prediction model, we measured the binding ability of PHEV Spike to the NCAM of different host. More obvious binding activity was observed to the rabbit receptor, rather than rat or human, suggesting that PHEV might be more adaptive to rabbit, which is obviously consistent with our prediction results. However, the measured binding ability in our experiments was weak, whereby we thought that only the RBD region of Spike protein might be the main influencing factor. The main active domain of Spike protein is the RBD, as confirmed in previous research. However, the actual binding ability may be affected by other regions of Spike, and the RBD may not represent the whole native conformation of Spike; therefore, only using the RBD might be one limitation in this study.

However, studies such as ours and those of others based on a public CoV sequence database are sensitive to the quality and distribution of available sequence data. In this study, the imbalance in the number of various types of samples had a certain impact on the results. The few available samples of ROD_LAG CoV led to an imbalance in the number of CoVs from different hosts. Some ROD_LAG samples were generated during oversampling, resulting in a relatively scattered distribution of these samples in unsupervised learning results and the full connection layer data of trained deep learning models. Therefore, a greater accumulation of CoV samples may help to improve the accuracy of the models. In addition, the different receptor binding ability of PHEV *Spike* protein with receptors in rat and rabbit indicates that a refinement of host classification might also be helpful to establish more accurate prediction models.

## 5. Conclusions

In summary, the DCR trait was confirmed to be representative of the CoV genome, and the DCR-based deep learning model worked well to assess the adaptation of Suiformes CoVs to other mammals. Suiformes might be intermediate hosts for human CoVs and other mammalian CoVs. The present study provides a novel approach to assess the risk of adaptation and transmission to humans and other mammals of Suiformes CoVs.

## Figures and Tables

**Figure 1 viruses-15-01556-f001:**
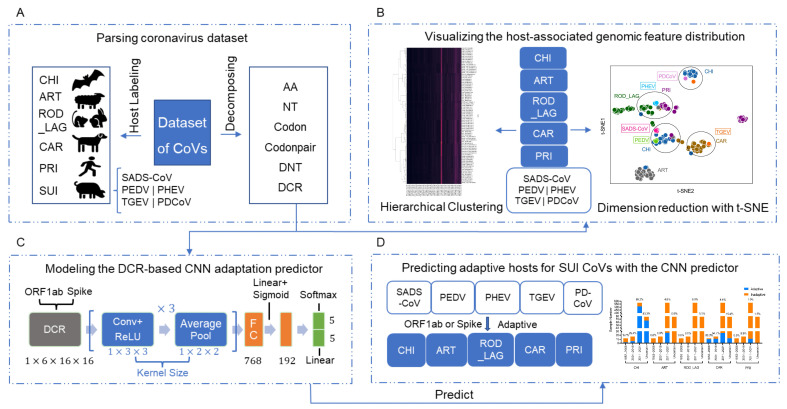
Pipeline of data processing, unsupervised learning, and prediction of adaptive hosts of SUI CoVs. The pipeline of this study can be divided into five parts. Host labeling (CHI, ART, ROD_LAG, CAR, PRI, and SUI (including SADS-CoV, PEDV, PHEV, TGEV, and PDCoV)), and *ORF1ab* and *Spike* sequences decomposition (six types of compositional traits: AAs, NTs, codons, codon pairs, DNTs, and DCRs) (**A**); reduction in and hierarchical clustering of each type of compositional trait and visualization (**B**); deep learning classification models established on the basis of DCR of *ORF1ab* and *Spike* sequences (**C**); prediction of adaptive hosts of each type of SUI CoV and temporal and spatial distributions of adaptation ratio (**D**).

**Figure 2 viruses-15-01556-f002:**
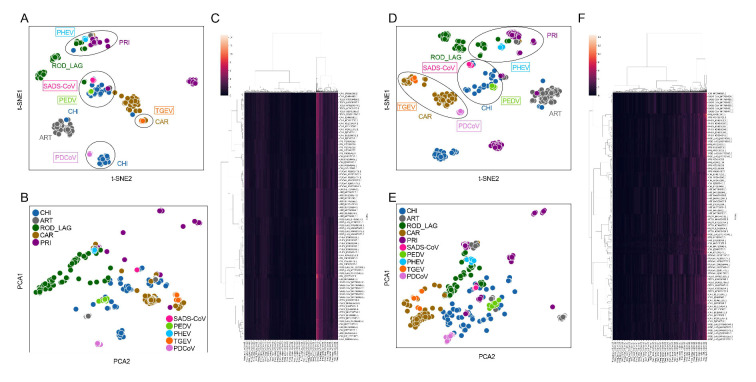
Reduction in, visualization of, and clustering of each type of SUI CoV and other CoVs based on DCR of *ORF1ab* and *Spike* sequences. Visualization of DCR reduced with t-SNE (**A**) and PCA (**B**), and hierarchical clustering of DCR of *ORF1ab* from each CoV sample (**C**); visualization of DCR reduced with t-SNE (**D**) and PCA (**E**), and hierarchical clustering of DCR of *Spike* from each CoV sample (**F**).

**Figure 3 viruses-15-01556-f003:**
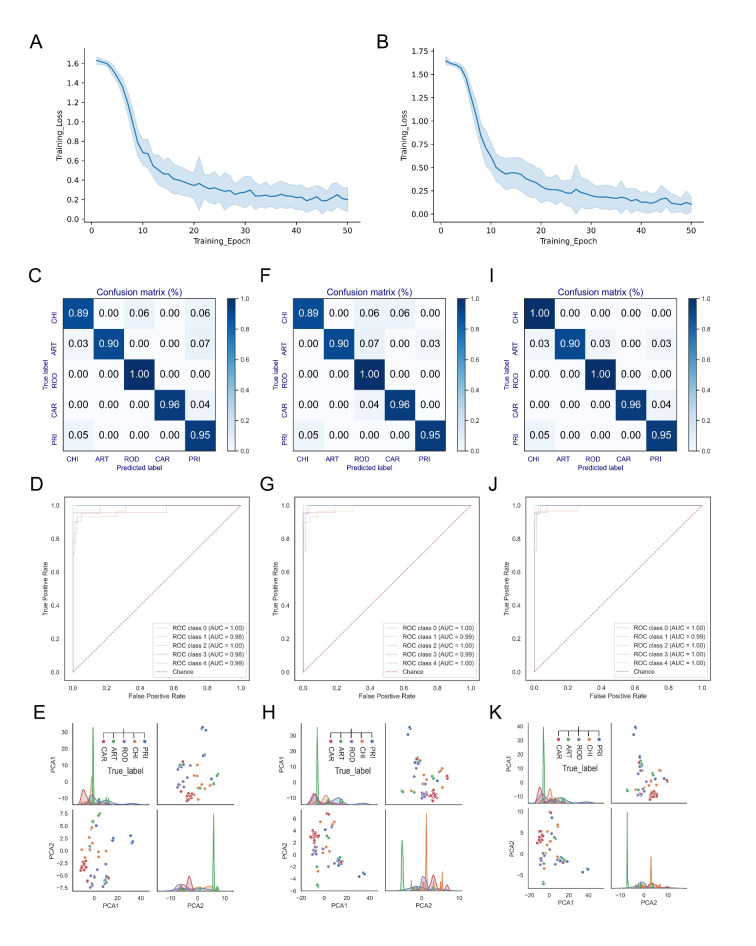
Performance of the DCR-based CNN models of *ORF1ab* and *Spike* gene. The training loss plotted with the average training loss value and the standard deviation of every training epoch (1–50) of the *ORF1ab* (**A**) and *Spike* (**B**) models, the confusion matrices (**C**), ROC with AUC (**D**), and the pair-plotting of PCA1 and PCA2 of the FC data (**E**) of the *Spike* model with training epochs of 10, 30 (**F**–**H**), and 50 (**I**–**K**). ROD means ROD_LAG.

**Figure 4 viruses-15-01556-f004:**
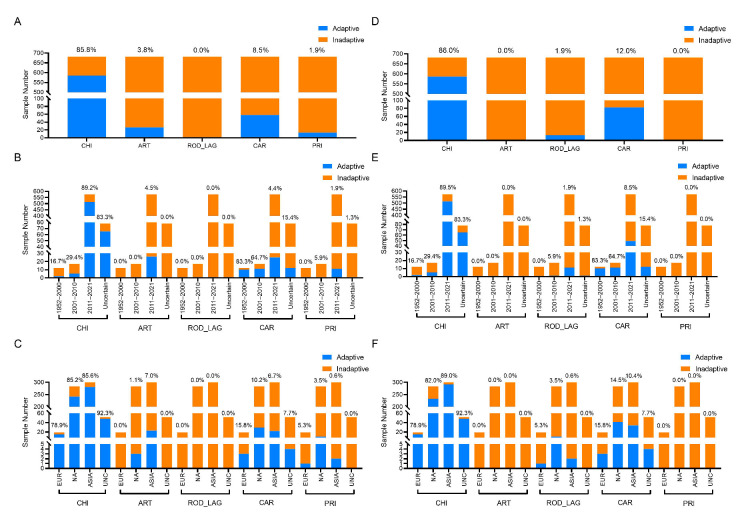
The adaptation ratio of SUI CoVs to various hosts and the spatiotemporal distribution of the adaptation ratio. The adaptation ratio of SUI CoV samples to CHI, ART, ROD_LAG, CAR, and PRI (**A**), the adaptation ratio of SUI CoV samples collected from 1952 to 2000, 2001 to 2010, and 2011 to 2021, as well as the samples with unknown collection date (**B**), and the adaptation ratio of SUI CoV samples collected from Europe (EUR), North America (NA), and Asia (ASIA), as well as the samples with unknown collection continent (UNC) (**C**), of the *ORF1ab* and *Spike* (**D**–**F**) models.

**Figure 5 viruses-15-01556-f005:**
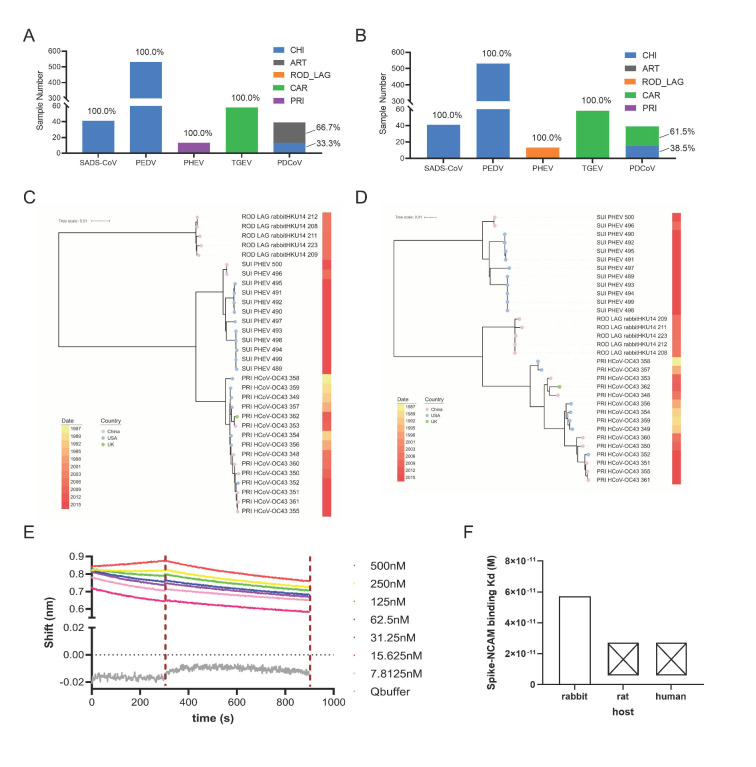
The adaptation prediction, phylogenetic analysis, and receptor binding verification of PHEVs, PRI, and ROD_LAG CoVs. The adaptation ratio of SUI CoVs to CHI, ART, ROD_LAG, CAR, and PRI of *ORF1ab* (**A**) or *Spike* gene (**B**). The phylogenetic tree was constructed using iqtree with 100 bootstrap replicates for randomly sampled PHEVs, PRI HCoV-OC43 CoVs, and ROD_LAG HKU14 CoVs, which were close in DCR distance to PHEVs for *ORF1ab* (**C**) and *Spike* (**D**) genes. Different colored circles in the tree represent different sample locations (pink: China, cyan: USA, green: UK). Sampling date is indicated by the progressive color bar at right. (**E**) Biolayer interferometry (BLI) assay for the interaction between RBD of the PHEV *Spike* protein and NCAM of rabbit. (**F**) The equilibrium dissociation constant (K_D_) of PHEV *Spike* protein with NCAM of rat, human, and rabbit.

## Data Availability

All original and cleaned CoV sequence data, and scripts for the project are available online from Github: https://github.com/Jamalijama/SwineCoVadaptation (accessed on 26 February 2023).

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
