# Peer review of "Risk Assessment of the Possible Intermediate Host Role of Pigs for Coronaviruses with a Deep Learning Predictor"

_viruses, 2023, doi:10.3390/v15071556_

Round 1

Reviewer 1 Report

In this manuscript, the authors establish deep learning models to predict host adaptation of swine CoVs. These studies reveal the potential intermediate host of swine CoVs and the transmission risk of CoVs to human. However, I have some major concerns and suggestions that need to be addressed.

1. The authors put the sequences of SADS-CoV, PEDV, PHEV and TGEV in the data processing, but PDCoV is also a major swine CoV causing diarrhea to pig in recent years. Please provide more data of PDCoV to the adaptive hosts.

2. The authors establish the models with CoV sequence data of different host species. Among them the genotype of CoV is important, such as HCoV-229E and PEDV belong to Alpha-CoV, HCoV-OC43, HCoV-HKU1 and PHEV belong to Beta-CoV, PDCoV belongs to Delta-CoV. The genotype needs to be analyzed and discussed in the CNN models.

3. In CoV studies, HCoV-OC43 was originated from cattle. Please check this in line 54.

4. Please analyze the abundance of coronavirus sequences among different species, this could affect the analysis of adaptation to different species.

5. It may be possible to provide some experimental evidence and references to verify the analysis results of the models.

6. There is a linguistic error in the article, “TGE” might be spelled “TGEV” in line 187. Please check others in this manuscript.

Author Response

Response to Reviewer #1

Reviewer #1

General: In this manuscript, the authors establish deep learning models to predict host adaptation of swine CoVs. These studies reveal the potential intermediate host of swine CoVs and the transmission risk of CoVs to human. However, I have some major concerns and suggestions that need to be addressed.

[Response]: We thank the reviewer for his or her kind words. Based on these comments and suggestions, we have made extensive modifications on the original manuscript.

Major Remarks

  1. The authors put the sequences of SADS-CoV, PEDV, PHEV and TGEV in the data processing, but PDCoV is also a major swine CoV causing diarrhea to pig in recent years. Please provide more data of PDCoV to the adaptive hosts.

[Response]: Thanks for the reviewer’s valuable suggestion. According to the reviewer’s suggestion, we have provided the host adaptability data of PDCoV in our original manuscript. 40 strains of PDCoV with full genome sequences was found and downloaded from NCBI nucleotide database (https://www.ncbi.nlm.nih.gov/nuccore). The related reference of some strains of PDCoV is as followed, “Lednicky J. A., et al. Independent infections of porcine deltacoronavirus among Haitian children. Nature. 2021.” We counted the frequency of 6 types of compositional traits (20 amino acids (AAs), 12 nucleotides (NTs), 48 dinucleotides (DNTs), 64 codons, 1536 dinucleotide composition representation (DCR) and 3721 codon pairs (codonpairs)) for ORF1ab or Spike sequence of each PDCoV sample (See Line 225-229), and analyzed the samples with unsupervised learning (See Line 256-279). We also predicted the host adaptation of these PDCoV samples by our deep learning models (See Line 377-381). Meanwhile, we have updated the figures containing the results of PDCoV.

  1. The authors establish the models with CoV sequence data of different host species. Among them the genotype of CoV is important, such as HCoV-229E and PEDV belong to Alpha-CoV, HCoV-OC43, HCoV-HKU1 and PHEV belong to Beta-CoV, PDCoV belongs to Delta-CoV. The genotype needs to be analyzed and discussed in the CNN models.

[Response]: Thanks for the reviewer’s valuable suggestion. As the reviewer mentioned, the genotype is an important feature in traditional biological evolution analysis.  According to the phylogenetic tree, coronaviruses can be divided into four different genotypes, Alpha-CoV, Beta-CoV, Gamma-CoV and Delta-CoV. Therefore, the genotype can well indicate the distance in the phylogenetic relationship. However, the present study aimed to build an alternative method to learning the general adaptability, regardless genotype. We focused on the host adaptability of swine coronaviruses based mainly on the dinucleotide composition representation (DCR) traits of ORF1ab and Spike sequence, independent of the genotype. We tried to establish a host adaptability prediction model which is completely different from the traditional genotyping methods. Based on above reasons, we do not include genotype analysis in the CNN models.

  1. In CoV studies, HCoV-OC43 was originated from cattle. Please check this in line 54.

[Response]: Thanks for pointing out this. We have read related articles carefully and conducted revision accordingly.

  1. Please analyze the abundance of coronavirus sequences among different species, this could affect the analysis of adaptation to different species.

[Response]: Thanks for the reviewer’s valuable suggestion. In our study, we did find that the sample size had a significant impact on the results. Considering that the sequences collected after December 31, 2019 contain a large number of SARS-CoV-2 sequences, to simplify data processing, we first downloaded coronavirus sequences collected as of December 31, 2019 and then added SARS-CoV-2 sequences and some SUI CoVs collected after 2019. The data related reference is as followed, “Li J., et al. Deep learning based on biologically interpretable genome representation predicts two types of human adaptation of SARS-CoV-2 variants. Brief. Bioinform. 2022.” After data cleaning and host labeling, the sample size for CHI, ART, ROD_LAG, CAR, PRI and SUI CoVs (including 41 SADS-CoV samples, 530 PEDV samples, 13 PHEV samples, 58 TGEV samples and 40 PDCoV samples) are 301, 579, 48, 79, 485 and 682, respectively. In unsupervised learning, sklearn.decomposition.PCA and sklearn.manifold.TSNE (https://scikit-learn.org/stable/about.html#citing-scikit-learn) in python were utilized to perform t-Distributed Stochastic Neighbor Embedding (t-SNE) and Principal Component Analysis (PCA) for dimensional reduction of 6 types of compositional traits of ORF1ab or Spike of 5 types of SUI CoVs and other CoVs. We found that unbalanced sample sizes for CoVs from different adaptive hosts may lead to an unclear separation and clustering of the samples. Therefore, we conducted downsampling and oversampling to reduce the imbalance of the sample sizes. We used pandas.DataFrame.sample of python to conduct downsampling, so as to reduce the sample size of SUI CoVs to 65 (SADS-CoV, PEDV, PHEV, TGEV and PDCoV each accounted for 20.0%). Considering the small number of PHEV samples in SUI CoVs, we did not performed oversampling to SUI samples in order to prevent the generated data from deviating from the original distribution. We also performed downsampling to reduce the sample size of CHI, ART, PRI CoVs to about 97 each and we used imblearn.over_sampling.SMOTE of python for oversampling to ensure the number of samples from CHI, ART, ROD_LAG, CAR and PRI is identical. After sampling, there were about 550 samples in total (CHI, ART, ROD_LAG, CAR and PRI CoVs each accounted for 17.6% and SUI CoVs accounted for 11.8%). We also conducted hierarchical clustering of each type of compositional trait with sns.clustermap of python to analyze the distance between SUI CoVs and other CoVs. To prevent samples from being hidden in hierarchical clustering results, we further extracted fewer samples from various types of CoVs. We performed downsampling to reduce samples of CHI, ART, ROD_LAG, CAR and PRI to 20 each, and samples of SUI to 65(SADS-CoV, PEDV, PHEV, TGEV and PDCoV each accounted for 20.0%). After sampling, there were 165 samples (CHI, ART, ROD_LAG, CAR and PRI CoVs each accounted for 12.1% and SUI CoVs accounted for 39.4%). The sampling of CHI, ART, ROD_LAG, CAR and PRI CoVs in supervised learning was consistent with sampling in dimensionality reduction of unsupervised learning. The specific descriptions have been added to the “ 2.2. section of Methods” (See Line 140-175).

  1. It may be possible to provide some experimental evidence and references to verify the analysis results of the models.

[Response]: Thanks for the reviewer’s valuable suggestion. Experimental verification is the most direct and effective method for the host adaptability study of the virus. Our study predicts the adaptability of Suiformes CoVs to 5 types of hosts, and scientifically assesses the infection risk of existing or novel Suiformes CoVs to various mammalian hosts, especially humans. The proliferation of different Suiformes CoVs in different mammalian host cells will be involved in experimental verification, which might cause high risk biosafety problems. Considering this, we didn’t perform experimental verification, but only supported by references.

  1. There is a linguistic error in the article,“TGE” might be spelled “TGEV” in line 187.Please check others in this manuscript.

[Response]: We have corrected it in Line 236 (previous Line 187) and checked the article throughout.

Reviewer 2 Report

Overall Comments

In this manuscript, Jiang et al. and co-authors aimed to establish 3D-CNN classification models based on viral genomic DCR to predict the adaptation of Suiformes CoVs to five types of hosts. They performed a detailed study to explore the potential of Suiformes as intermediate hosts in CoV transmission from their natural hosts to humans. They found that Suiformes might be intermediate hosts for human CoVs and other mammalian CoVs. The present study provides a novel approach to assess the risk of adaptation and transmission to humans and other mammals of swine CoVs. The deep learning predictor used in this study can help identify potential intermediate hosts for emerging viruses, which can aid in early detection, prevention, and control measures.

Overall, the approach is interesting and no other study is available in this area. Authors should be more careful with the language and precise. However, several points need to be addressed before the manuscript can be recommended for publication.

Points need to be improved:

1) The Introduction section was not well-written. The authors should introduce the methods used to predict coronavirus host adaptability in more detail and include references to recent literature on this topic.

2) Swine coronaviruses (CoVs) encode four structural proteins, as well as multiple non-structural and accessory proteins. In this study, the authors developed Convolutional Neural Networks (CNN) models to predict host adaptation of swine CoVs, using the sequence of the ORF1ab and Spike proteins. However, it is unclear why the authors focused on these two proteins and not on other structural proteins like N or M proteins or accessory proteins like ORF3 or ORF7. Therefore, the authors should explain in detail in the background introduction section the criteria and rationale for this decision.

3) The use of Convolutional Neural Networks (CNN) models to predict host adaptation of swine CoVs is a new field of research. Is there any relevant research on the host adaptability of human coronaviruses using CNN models? Please introduce in the introduction section.

4) Previous research has investigated the host adaptability of swine coronaviruses using codon usage bias and other methods (10.1371/journal.ppat.1009149; 10.3389/fmicb.2021.738082; 10.1038/msb.2009.71), what’s the main differences between those? In the discussion section, the authors should refer to this research and compare the characteristics of these two approaches, highlighting their main differences. By doing so, the authors can provide a comprehensive analysis of the advantages and limitations of each method, and how their own study adds to the existing knowledge in the field.

Author Response

Response to Reviewer #2

Reviewer #2

General: In this manuscript, Jiang et al. and co-authors aimed to establish 3D-CNN classification models based on viral genomic DCR to predict the adaptation of Suiformes CoVs to five types of hosts. They performed a detailed study to explore the potential of Suiformes as intermediate hosts in CoV transmission from their natural hosts to humans. They found that Suiformes might be intermediate hosts for human CoVs and other mammalian CoVs. The present study provides a novel approach to assess the risk of adaptation and transmission to humans and other mammals of swine CoVs. The deep learning predictor used in this study can help identify potential intermediate hosts for emerging viruses, which can aid in early detection, prevention, and control measures.

Overall, the approach is interesting and no other study is available in this area. Authors should be more careful with the language and precise. However, several points need to be addressed before the manuscript can be recommended for publication.

[Response]: We thank for the reviewer’s good evaluation and kind suggestion. We have revised the manuscript accordingly.

  1. The Introduction section was not well-written. The authors should introduce the methods used to predict coronavirus host adaptability in more detail and include references to recent literature on this topic.

[Response]: Thanks for the reviewer’s professional comments. In accordance with the suggestions, we have added some details of the prediction model for virus host adaptability based on genome composition analysis, and cited 12 new references (See Line 75-85, Line 87-97).

  1. Swine coronaviruses (CoVs) encode four structural proteins, as well as multiple non-structural and accessory proteins. In this study, the authors developed Convolutional Neural Networks (CNN) models to predict host adaptation of swine CoVs, using the sequence of the ORF1ab and Spike proteins. However, it is unclear why the authors focused on these two proteins and not on other structural proteins like N or M proteins or accessory proteins like ORF3 or ORF7. Therefore, the authors should explain in detail in the background introduction section the criteria and rationale for this decision.

[Response]: Thanks for the reviewer’s valuable suggestion. The ORF1ab and Spike sequences were selected for two main reasons. Firstly, the binding ability to specific receptor and the replication ability in host cells are considered to be the key factors affecting viral host adaptation. So that these two major viral proteins—receptor-binding glycoprotein (Gp; also named as S for CoVs) and RNA-dependent RNA polymerase (RdRp; mainly ORF1ab encoded) were assessed by the classification traits in our study. Second, among the coronavirus sequences we downloaded, the sequences of ORF1ab and Spike are relatively complete. The first reason has been supplemented in the Introduction section (See Line101-106). Additionally, other ORFs were not unanimously presented in all CoVs, which were analyzed in the present stud; E, M, N genes, which were unanimously presented in all analyzed CoVs, were not significantly separated among the six host groups, and clustered within same host group (Such part of results were presented in another manuscript, which were in press).

  1. The use of Convolutional Neural Networks (CNN) models to predict host adaptation of swine CoVs is a new field of research. Is there any relevant research on the host adaptability of human coronaviruses using CNN models? Please introduce in the introduction section.

[Response]: Thanks for the reviewer’s valuable suggestion. We have added descriptions in the introduction section about the host adaptability of human coronaviruses using CNN models (See Line 92-97), and related references have been cited.

  1. Previous research has investigated the host adaptability of swine coronaviruses using codon usage bias and other methods (10.1371/journal.ppat.1009149; 10.3389/fmicb.2021.738082; 10.1038/msb.2009.71), what’s the main differences between those? In the discussion section, the authors should refer to this research and compare the characteristics of these two approaches, highlighting their main differences. By doing so, the authors can provide a comprehensive analysis of the advantages and limitations of each method, and how their own study adds to the existing knowledge in the field.

[Response]: Thanks for reviewer’s valuable suggestions and related refs. We have learned all of these articles and added related descriptions in discussion section (See Line 421-434).

Reviewer 3 Report

The aim of this study was to determine is various Coronaviruses known to infect swine could potentially be zoonotic with the ability to adapt and spread to humans. Are swine potential intermediate hosts?  Comparing available sequences from several swine coronaviruses in a public genetic database, the authors determined the following:

1.  Swine acute diarrhea syndrome and porcine epidemic diarrhea virus are adapted to the Chiroptera species (bats); this is an interesting result as the origin of PEDV has not been determined.

2.  TGEV is adapted to Carnivora; not surprising since TGEV is related to canine coronavirus.

3.  HEV might be adapted to primate, Rodent or Lagomorpha species

Considering the host adaptability of other coronaviruses, perhaps these results should not be surprising, but the authors CNN model and unsupervised leaning analysis provide a potential tool for predicting potential zoonotic potential of swine coronaviruses.  The authors also provide evidence from literature that PDCoV may infect malnourished children in Haitia.  Authors have you tried to infect human cell lines with any of these swine viruses/

The study is based on the CNN model and there is not data presented that ny of these swine coronaviruses can or have been found to infect other hosts.  However, there is potential and this type of analysis could be beneficial to predict future pandemics?

Overall the paper is well presented.

Author Response

Response to Reviewer #3

Reviewer #3

General: The aim of this study was to determine is various Coronaviruses known to infect swine could potentially be zoonotic with the ability to adapt and spread to humans. Are swine potential intermediate hosts?  Comparing available sequences from several swine coronaviruses in a public genetic database, the authors determined the following:

  1. Swine acute diarrhea syndrome and porcine epidemic diarrhea virus are adapted to the Chiroptera species (bats); this is an interesting result as the origin of PEDV has not been determined.
  2. TGEV is adapted to Carnivora; not surprising since TGEV is related to canine coronavirus.
  3. HEV might be adapted to primate, Rodent or Lagomorpha species

[Response]: Thanks for reviewer’s approval of our research.

Considering the host adaptability of other coronaviruses, perhaps these results should not be surprising, but the authors CNN model and unsupervised leaning analysis provide a potential tool for predicting potential zoonotic potential of swine coronaviruses.  The authors also provide evidence from literature that PDCoV may infect malnourished children in Haitia.  Authors have you tried to infect human cell lines with any of these swine viruses?

[Response]: Thanks for reviewer’s suggestion. Infecting human cell lines with different swine coronaviruses will be helpful in reflecting different host adaptation. However, considering the experiment would have high-risk biosafety problems, we didn’t perform experimental verification.

The study is based on the CNN model and there is not data presented that any of these swine coronaviruses can or have been found to infect other hosts.  However, there is potential and this type of analysis could be beneficial to predict future pandemics?

[Response]: Thank for reviewer’s comments. Sequence compositions of nucleic acids and proteins are significantly associated with genome evolution and adaptation across all kingdoms of life. Adaptive determinants have recently been widely identified at the nucleic acid level (genomic DNA, RNA or mRNA) among pathogens such as parasites, bacteria and viruses. The dynamic homeostasis of genomic RNA sequences shapes the transcription, translation and decay of mRNA, particularly for RNA viruses. These determinants regulate the replication of pathogens in hosts via the machinery related to codon usage bias, the dinucleotide composition, tRNA abundance, mRNA decay, the translation elongation speed and translation efficiency. Thus, the RNA sequence-based nucleotide composition is biologically meaningful and is closely related to the causal inference of virus phenotype. Machine learning methods can efficiently learn the characteristics of nucleotide composition and then reflect the host adaptive phenotype. Machine learning models based on compositional traits such as codon, codon pair (codonpair), and dinucleotide (DNT) have enabled accurate prediction of virus adaptation to hosts. Deep learning models based on dinucleotide composition representation (DCR) and AA have also been utilized to predict the human adaptation of the SARS-CoV-2 variants. So this type of analysis could be great potential and beneficial to predict future pandemics.

Round 2

Reviewer 1 Report

The authors successfully responded to the comments and updated the manuscript as well. While, I think the authors can simply evaluate the results of the analysis by the interaction of the expressed viral protein (especially the spike protein) with cell lines of different species, which will not involve the issue of biosafety.

Author Response

The authors successfully responded to the comments and updated the manuscript as well. While, I think the authors can simply evaluate the results of the analysis by the interaction of the expressed viral protein (especially the spike protein) with cell lines of different species, which will not involve the issue of biosafety.

Response: Thanks for the reviewer’s professional comments. According to the suggestions, we selected PHEV to analyze the interaction between the spike protein with cell lines of human, rat and rabbit. According to our investigation, we determined that a fragment (258-amino-acid fragment, residues 291-548) located in Spike 277-794 was the RBD of the PHEV spike protein, and a cellular neural cell adhesion molecule (NCAM) was the key receptor protein (Dong B, et al. Intervirology. 2015; Gao W, et al. Virol J. 2010.). Then BLI binding experiments were performed to measure the binding ability of PHEV spike RBD to NCAM of different host cells. We found that PHEV spike RBD could bind weakly to rabbit NCAM protein, while could hardly bind to human or rat NCAM protein (Fig. 5E, F and Supplementary Fig. S6A-C). It meant the PHEV spike protein was more adaptive to rabbit, rather than rat or human, which was basically consistent with our prediction results that 100% of PHEVs were predicted to be adaptive to ROD_LAG for Spike genes (Fig. 5B). The relevant descriptions had been added in Methods (See Line 229-238), Results (See Line 408-422) and Discussion (See Line 474-484 and Line 493-495) in our revised manuscript.

Reviewer 2 Report

The author has adequately addressed all of the questions raised by the reviewers and made significant revisions, which warrant reconsideration for publication. 

Author Response

The author has adequately addressed all of the questions raised by the reviewers and made significant revisions, which warrant reconsideration for publication.

Response: Thanks for the reviewer’s approval.

Reviewer 3 Report

Authors thanks for your comments.  The use of human cell lines to determine if any of these coronaviruses could replicate in human cells could be high risk, but I think you could control the risk.  So this may be a future experimental approach.

Thanks for your comments.

Author Response

Authors thanks for your comments.  The use of human cell lines to determine if any of these coronaviruses could replicate in human cells could be high risk, but I think you could control the risk.  So this may be a future experimental approach.

Thanks for your comments.

Response: Thanks for the reviewer’s advices. To avoid the possible high-risk biosafety problems, we measured binding ability of Cov spike protein to different host receptors instead of using live virus. We selected PHEV to analyze the interaction between its spike protein with cell lines of human, rat and rabbit. According to our investigation, we determined that a fragment (258-amino-acid fragment, residues 291-548) located in Spike 277-794 was the RBD of the PHEV spike protein, and a cellular neural cell adhesion molecule (NCAM) was the key receptor protein (Dong B, et al. Intervirology. 2015; Gao W, et al. Virol J. 2010.). Then BLI binding experiments were performed to measure the binding ability of PHEV spike RBD to NCAM of different host cells. We found that PHEV spike RBD could bind weakly to rabbit NCAM protein, while could hardly bind to human or rat NCAM protein (Fig. 5E, F, Supplementary Fig. S6A-C). It meant the PHEV spike protein was more adaptive to rabbit, rather than rat or human, which was basically consistent with our prediction results that 100% of PHEVs were predicted to be adaptive to ROD_LAG for Spike genes (Fig. 5B). The relevant descriptions had been added in Methods (See Line 229-238), Results (See Line 408-422) and Discussion (See Line 474-484 and Line 493-495) in our revised manuscript. As you said, we will also look for opportunities to verify these with live coronaviruses in the future.